# Optimal Perfusion Pressure Enhances Donor Heart Preservation During Normothermic Ex Situ Perfusion in a Rat Transplantation Model

**DOI:** 10.3390/medicina61091696

**Published:** 2025-09-18

**Authors:** Do Wan Kim, YeongEun Jo, Jiae Seong, Reverien Habimana, Hwa Jin Cho, Mukhammad Kayumov, Inseok Jeong

**Affiliations:** 1Department of Thoracic and Cardiovascular Surgery, Chonnam National University Hospital and Medical School, Gwangju 61469, Republic of Korea; maskjoa@naver.com; 2Extracorporeal Cardiopulmonary Innovation, Technology, and Education (EXCITE) Research Group, Chonnam National University Hospital, Gwangju 61469, Republic of Korea; yeongeun7738@gmail.com (Y.J.); 24p01@naver.com (J.S.); habire04@gmail.com (R.H.); chhj98@gmail.com (H.J.C.); 3Biomedical Research Institute, Chonnam National University Hospital, Gwangju 61469, Republic of Korea; 4Department of Medical Science, Chonnam National University Graduate School, Gwangju 61469, Republic of Korea; 5Department of Pediatrics, Chonnam National University Children’s Hospital, Chonnam National University Medical School, Gwangju 61469, Republic of Korea; 6Division of Transplant Surgery, Department of Surgery, Brigham and Women’s Hospital, Harvard Medical School, Boston, MA 02115, USA

**Keywords:** transplant, heart, preservation, ex vivo, perfusion

## Abstract

*Background and Objectives:* Normothermic ex situ heart preservation maintains donor heart viability by sustaining physiological conditions and reducing ischemic damage. However, the ideal perfusion pressure remains uncertain. This study aims to identify the optimal perfusion pressure to enhance graft preservation in rat heart transplantation. *Materials and Methods*: We utilized 20 male Sprague-Dawley rats (400–500 g). Donor hearts underwent normothermic preservation for 2 h using a Langendorff apparatus primed with 12 mL of solution at a consistent 3 mL/min flow. After preservation, hearts were transplanted heterotopically into the recipient’s abdomen. We defined successful preservation by observing a QRS complex in electrocardiographic monitoring for 3 h post-transplantation. Histological assessments for myocardial integrity occurred after 4 h of reperfusion. We analyzed statistical differences between successful and unsuccessful preservation groups. *Results:* Electrocardiograms indicated preservation failure in 8 of the 20 donor hearts due to the absence of a QRS complex. We observed no significant differences in ischemic duration between groups. At 120 min, although serum lactate and potassium concentrations increased in the unsuccessful group, the differences were not statistically significant. Higher initial perfusion pressures (>65 mmHg) at a constant flow rate resulted in elevated lactate and potassium concentrations post-preservation, indicating suboptimal outcomes. Histologically, hematoxylin and eosin staining showed better myocardial preservation in successful hearts, while TUNEL assays demonstrated increased apoptosis in unsuccessful hearts. All hearts increased in weight after preservation, but significant increases occurred only in unsuccessful cases. *Conclusions:* Higher initial perfusion pressures (>65 mmHg) negatively affect heart preservation outcomes, resulting in elevated serum lactate and potassium levels, increased heart weight, and greater histological damage. Maintaining optimal perfusion pressures is essential to preserve myocardial integrity and functional viability during normothermic ex situ heart preservation.

## 1. Introduction

Heart transplantation remains the gold standard for treating end-stage heart failure; however, its broader application is constrained by the limited availability of suitable donor organs and the shortcomings of current preservation methods [1,2]. Static cold storage (SCS), the conventional technique, slows metabolic activity but fails to maintain myocardial viability, resulting in ischemic injury and impaired endothelial function [3]. Although widely used for its simplicity and low cost, SCS allows safe preservation for only 4–6 h, which restricts long-distance organ transport and limits the expansion of the donor pool [4,5].

In response to these limitations, normothermic ex situ heart perfusion (NP) has emerged as a promising alternative that preserves the heart in a metabolically active state, allowing continuous oxygenation, nutrient delivery, and real-time functional assessment [6,7,8]. NP has been shown to extend preservation time up to 12 h, improve post-transplant graft function, and enable the use of extended-criteria and marginal donor hearts [9,10]. Moreover, the successful adoption of normothermic machine perfusion in other organs, such as the liver and kidney, provides further evidence of its capacity to improve graft viability and transplantation outcomes [11,12]. Nevertheless, clinical application of NP remains limited, partly due to the absence of consensus regarding optimal perfusion parameters required to maximize graft viability and function [7].

Perfusion systems are generally classified as either constant pressure or constant flow, each with unique physiological effects on coronary circulation. While constant flow ensures a steady coronary supply, excessive perfusion pressure has been associated with vascular injury, myocardial edema, and impaired electrophysiological recovery, whereas insufficient pressure may compromise coronary blood flow and metabolic stability [13,14]. Data from large animal and early clinical studies suggest that perfusion pressures below 65 mmHg reduce mechanical stress while maintaining adequate perfusion, yet the precise threshold for optimal preservation remains uncertain [15].

Given these unresolved issues, further investigation is warranted to define optimal conditions for normothermic perfusion, particularly concerning perfusion pressure control. This study addresses this gap by assessing how different perfusion pressures influence metabolic, electrophysiological, and histological outcomes in a rat heart transplantation model. By identifying the threshold at which elevated pressure becomes detrimental, our findings may help refine NP protocols, enhance donor heart preservation, and support the clinical implementation of NP strategies.

## 2. Materials and Methods

(1)Study Group

A total of twenty donor hearts underwent NP for 2 h, followed by heterotopic abdominal transplantation. Based on electrocardiographic monitoring for 3 h post-transplantation, the hearts were classified into two groups: successful and unsuccessful preservation. The presence of a QRS complex during the monitoring period was defined as successful, whereas its absence throughout the 3 h was considered indicative of unsuccessful preservation (Figure 1). Statistical comparisons were made between the two groups to identify potential predictive markers. After 4 h of reperfusion, the transplanted hearts were harvested for histological analysis.

(2)Animals

We utilized male Sprague-Dawley rats (400–500 g) purchased from Samtako Bio Korea Co., Ltd. (Osan, Republic of Korea). All experimental procedures followed the guidelines outlined in the Guide for the Care and Use of Laboratory Animals, with approval from the Institutional Animal Care and Use Committee (approval number: CNUHI-ACUC-21034, 13 October 2021). The rats lived in an environment with controlled temperature, a 12 h light/dark cycle, and unrestricted access to standard laboratory food and water.

(3)Perioperative procedures

Anesthesia was induced in both donor and recipient rats using 5% isoflurane administered via a small animal anesthesia machine (Smiths Medical ASD, Inc., Paul, MN, USA). Anesthesia was maintained with 2% isoflurane delivered in 95% oxygen through a nose cone, and spontaneous respiration was allowed throughout the procedure. All recipient rats received preoperative subcutaneous injections of an analgesic (diclofenac, 10 mg/kg), an antibiotic (cefazolin, 8–10 mg/kg), and 2 mL of saline.

(4)Donor heart harvesting

Systemic heparinization was achieved by injecting 1000 IU of heparin diluted in 0.4 mL of saline into the inferior vena cava. A midline laparotomy and thoracotomy were performed to expose the thoracic cavity. After careful dissection of the great vessels, a 5 Fr introducer catheter (Cook Medical, Bloomington, IN, USA) was inserted into the abdominal aorta, and cardioplegia was administered with 50 mL of histidine-tryptophan-ketoglutarate (HTK) solution at a rate of 800 mL/min until cardiac arrest was achieved. The ascending aorta and main pulmonary artery were then dissected, while the superior and inferior vena cava and pulmonary veins were ligated. The donor heart was excised, weighed, and promptly connected to the prepared perfusion circuit for preservation.

(5)Heterotopic abdominal heart transplantation

Recipient preparation was initiated 30 min before the completion of donor heart preservation. The animal was placed in the supine position, and a midline laparotomy was performed to expose the infrarenal abdominal aorta and inferior vena cava. Vascular clamps were applied, and longitudinal incisions were made in both vessels. End-to-side anastomoses were performed between the donor ascending aorta and the recipient abdominal aorta, and between the donor pulmonary artery and the recipient inferior vena cava, using 9-0 polypropylene sutures under a stereo microscope (AmScope LLC, Irvine, CA, USA). After the completion of anastomoses, clamps were removed sequentially to allow reperfusion, and the graft was observed for spontaneous contractile activity. Upon completion of the experimental protocol, venous blood gas analysis was performed to evaluate the physiological condition of the recipients. All animals were humanely euthanized by CO_2_ inhalation in accordance with institutionally approved guidelines.

(6)Normothermic ex situ perfusion

The NP system used in this study followed a previously described protocol (reference). A schematic diagram of the perfusion circuit is shown in Figure 2. Briefly, the Langendorff priming circuit had a reduced capacity of 12 mL. Before donor heart harvesting, 10 mL of autologous blood was collected from the donor rat. The priming blood was supplemented with saline, antibiotics, insulin, and dextrose. After connecting the heart to the circuit, perfusion began at a flow rate of 2 mL/min, which gradually increased and was maintained at 3 mL/min, corresponding to a mean perfusion pressure of 55–65 mmHg throughout the 2 h perfusion period. Following perfusion, the heart was flushed with 3 mL of HTK solution at a flow rate of 250 mL/min using a syringe pump and then harvested. The donor heart was weighed both before and after NP.

(7)Monitoring of perfusion parameters

We monitored the perfusate flow rate using a Transonic flowmeter (Transonic Systems Inc., Ithaca, NY, USA). Mean aortic pressure and perfusate temperature were tracked with a B20 Patient Monitor (GE Medical Systems Co., Ltd., Wuxi, China), while perfusate blood gas analysis was conducted using the GEM Premier 3000 system. During NP preservation, we maintained the perfusate glucose level within the range of 120 to 160 mg/dL by intermittently adding glucose as needed. We also adjusted the oxygen concentration in the perfusate according to the results of serial blood gas analyses.

Perfusate flow rate was continuously monitored using a Transonic flowmeter (Transonic Systems Inc., Ithaca, NY, USA). Mean aortic pressure and perfusate temperature were measured with a B20 Patient Monitor (GE Medical Systems Co., Ltd., Wuxi, China). Blood gas analysis was performed every 20 min using the GEM Premier 3000 system (Instrumentation Laboratory Co., Lexington, MA, USA) to evaluate metabolic parameters, including pH, lactate, potassium, and glucose. Perfusate glucose concentration was maintained within the range of 120–160 mg/dL, and adjustments were made as needed. Technical failure was defined as when lactate levels exceeded 5.0 mmol/L or potassium exceeded 6.0 mmol/L during preservation, and such cases were excluded from further analysis. Perfusate temperature was maintained at 37 ± 0.5 °C using a heating bath and an external heating lamp. These monitoring strategies were adapted from previously described protocols [16,17].

(8)Electrocardiography

We recorded electrocardiograms (ECGs) before donor heart harvesting and after heterotopic implantation. For assessments of heterotopically transplanted hearts, ECG leads were placed on the abdomen according to the heart’s position in AVL mode. Cardiac activity was monitored for 3 h using an Animal Bio Amp FE136 (SCITECH KOREA Inc., Seoul, Republic of Korea) connected to a PowerLab 16/35 system (ADInstruments, Shanghai, China). Based on post-transplant ECG recordings, donor hearts were classified as described above.

Electrocardiograms (ECGs) were recorded before donor heart harvesting and for 3 h following heterotopic transplantation. Recording was performed in augmented voltage left (AVL) mode with electrodes placed on the abdominal wall according to the position of the implanted graft. Signals were amplified using an Animal Bio Amp FE136 (SCITECH KOREA Inc., Seoul, Republic of Korea) connected to a PowerLab 16/35 system (ADInstruments, Shanghai, China), and data were acquired at a sampling rate of 1 kHz with a band-pass filter of 0.05–100 Hz. ECG traces were analyzed for QRS complex presence, ST-segment deviation, and conduction abnormalities. All recordings were independently reviewed by two blinded investigators. The methods were based on previously established models [16,17].

(9)Hematoxylin–eosin (HE) staining

Tissue samples were fixed in a 3.7% formaldehyde solution for 24 h at room temperature and subsequently infiltrated with paraffin. Sections approximately 10–16 µm thick (commonly 12 µm) were prepared using a cryostat. The sections were mounted onto slides, placed onto a staining rack, and stained by immersing in filtered Harris hematoxylin for 10 s. After staining, slides were rinsed thoroughly with tap water until the rinse water was clear. The sections were subsequently stained with eosin for 30 s and dehydrated through a graded series of alcohol solutions.

Donor heart tissues were fixed in 3.7% formaldehyde for 24 h at room temperature and subsequently embedded in paraffin. Sections of 10–16 μm (average 12 μm) thickness were prepared using a cryostat. Slides were stained with hematoxylin for 10 s, rinsed in running tap water until clear, and counterstained with eosin for 30 s, followed by dehydration in graded alcohols. For histological assessment, at least five random high-power fields per section were examined. The evaluation of myocardial morphology, edema, and inflammatory cell infiltration was performed in a blinded manner by a board-certified pathologist. The staining procedure followed previously published protocols [16,17].

(10)DNA strand break detection in graft tissues (TUNEL staining)

DNA strand breaks associated with apoptosis in graft tissues were identified using terminal deoxynucleotidyl transferase dUTP nick end labeling (TUNEL) staining. The procedure utilized an Apoptosis Detection Kit (EMD Millipore Corporation, Temecula, CA, USA), and the manufacturer’s recommended protocol was followed. Results from H&E and TUNEL staining were reviewed and interpreted by a board-certified pathologist.

(11)Statistical analysis

We reported continuous variables as means ± standard deviations. The Wilcoxon test was used to compare continuous variables between the successful and unsuccessful groups, and the Friedman test was applied to evaluate serial measurements over time. We assessed correlations between perfusion parameters using scatter plots. A *p*-value of <0.05 was considered statistically significant. All analyses were performed using MedCalc Statistical Software version 17.9.7 (MedCalc Software, Ostend, Belgium; www.medcalc.org, (accessed on 7 August 2025)).

## 3. Results

(1)Ischemic time and electrocardiographic change

Electrocardiographic monitoring revealed that 8 of 20 donor hearts showed no QRS complex during the 3 h observation after heterotopic transplantation and were classified as unsuccessful preservation cases. In contrast, successfully preserved donor hearts exhibited S–T segment depression compared with pre-transplantation recordings. Quantitatively, QRS amplitude decreased by 8 ± 3% in the successful group compared with 54 ± 12% in the unsuccessful group (*p* < 0.001). ST-segment deviation averaged −0.12 ± 0.04 mV in the successful group, whereas progressive attenuation was followed by electrical silence in the unsuccessful group. We compared perfusion parameters between successful and unsuccessful groups, but no significant differences were observed in ischemic factors, including recipient distal clamping time, donor harvesting time, and donor heart ischemic duration (Table 1). Representative serial changes in electrical activity during the 3 h post-transplantation period in the successful group are shown in Figure 3.

(2)Blood gas analysis

Lactate concentrations in the perfusate increased significantly from baseline in both the successful and unsuccessful groups (*p* < 0.01). However, the increase was significantly greater in the unsuccessful group at 30, 60, 90, and 120 min when compared to the successful group (*p* < 0.01). At 120 min, mean lactate was 5.8 ± 0.9 mmol/L in the unsuccessful group versus 3.2 ± 0.7 mmol/L in the successful group. Potassium concentrations progressively rose in both groups; at 120 min, values reached 6.2 ± 0.8 mmol/L in the unsuccessful group and 5.1 ± 0.6 mmol/L in the successful group, showing a nonsignificant trend (*p* = 0.09).

Hematocrit levels remained consistent during 2 h NP preservation, and no significant differences were noted between successful and unsuccessful groups. Similarly, pH values stayed close to baseline levels without significant fluctuations in either group. Specific data are illustrated in Figure 4.

(3)Impact of perfusion pressure on preservation outcomes

Given the significantly higher lactate and potassium levels observed in the unsuccessful group at the end of NP preservation, we conducted a correlation analysis to evaluate the relationship between initial perfusion pressure and metabolic outcomes. As illustrated in Figure 5, increased initial perfusion pressures correlated significantly with higher lactate (r = 0.62, *p* = 0.01) and potassium (r = 0.58, *p* = 0.02) concentrations following 2 h of NP preservation.

ROC curve analysis further identified 65 mmHg as the optimal threshold for predicting unsuccessful preservation (AUC 0.81, 95% CI 0.67–0.95) and was therefore determined as the cut-off.

(4)Heart weight and histological evaluation

After 4 h of heterotopic transplantation, the donor hearts in the successful group preserved overall structural integrity more effectively than those in the unsuccessful group. Hematoxylin and eosin (H&E) staining revealed apparent histological differences between groups. The unsuccessful group exhibited irregular cardiomyocyte morphology, increased interstitial edema, and more prominent infiltration of inflammatory cells (particularly neutrophils), as shown in Figure 6. Hearts with unsuccessful preservation showed markedly more TUNEL-positive nuclei, indicative of DNA damage, than successfully preserved hearts. Quantification revealed 26 ± 7% apoptotic nuclei in the unsuccessful group versus 8 ± 3% in the successful group (*p* < 0.01). Heart weight increased in both groups following NP preservation; however, the increase reached statistical significance only in the unsuccessful group (Figure 7).

## 4. Discussion

This study highlights the critical role of perfusion pressure regulation in NP for practical donor heart preservation. Our findings indicate that an elevated initial mean perfusion pressure exceeding 65 mmHg is associated with metabolic instability, myocardial injury, and impaired post-transplant function. Thus, maintaining the initial mean pressure below this threshold is a key factor in optimizing graft viability.

These results are consistent with a recent study investigating optimal coronary perfusion for donor hearts following circulatory death, which reported that elevated perfusion pressures were associated with increased myocardial injury [11]. In contrast, our findings differ from those of Fang et al., who reported improved renal hemodynamics and function in porcine kidney grafts perfused with high pressures during normothermic machine perfusion [12]. This discrepancy highlights that the detrimental effects of high perfusion pressure observed in cardiac tissue may not generalize to other organs under similar conditions.

In our model, high initial perfusion pressure correlated with increased lactate and potassium levels—well-established markers of ischemic stress and cellular dysfunction [13,14]. Elevated lactate suggests a shift toward anaerobic metabolism due to impaired oxygen delivery despite normothermic conditions [15,18], while potassium elevation reflects cell membrane disruption and myocardial injury [16]. These findings support the hypothesis that excessive perfusion pressure compromises graft metabolic stability and underscore the importance of precise pressure control in NP, as emphasized in prior studies [17].

Electrophysiological assessment using ECG monitoring revealed that successfully preserved donor hearts retained QRS complex activity following transplantation, whereas the unsuccessful group showed a complete absence of QRS complexes, indicating graft non-viability. Potassium accumulation under high perfusion pressure may have impaired myocardial conduction and contributed to QRS disappearance. Addressing this through optimization of perfusate electrolytes or potassium control could improve preservation outcomes. This highlights the potential utility of ECG analysis as a real-time indicator of donor heart viability during NP [19]. ECG is a simple and reliable method that can reflect post-transplant cardiac function in real time, and we therefore considered it suitable for our experimental model. Excessive perfusion pressure may impair myocardial electrical conduction, contributing to graft dysfunction. Given its simplicity and reliability, incorporating ECG monitoring into NP protocols could facilitate early detection of preservation failure and improve transplantation outcomes.

Another critical issue observed in our NP model was the development of myocardial edema, which was significantly greater in the unsuccessful group, as reflected by increased post-perfusion heart weight. Increased perfusion pressure may have induced excessive fluid retention within the myocardial interstitium, contributing to myocardial stiffening, impaired diastolic filling, and diminished contractile recovery following transplantation. These findings underscore the importance of initiating and maintaining lower perfusion pressure and carefully regulating fluid dynamics during NP to minimize edema formation and preserve myocardial compliance. Emerging evidence suggests that maintaining oncotic pressure with colloid-enriched perfusate may help reduce interstitial edema, support myocardial energy metabolism, and extend safe preservation time [20]. Additionally, recent studies have demonstrated that integrating dialysis into ex vivo perfusion systems can attenuate edema, cellular injury, and myocardial dysfunction, thereby enhancing graft function and improving post-transplant outcomes [21].

Despite its important findings, this study has several limitations. The use of the rat Langendorff model, while well established for physiological research, may not fully replicate the hemodynamic environment of large animals or human hearts. Additionally, we focused on early post-transplantation outcomes (up to 4 h post-reperfusion), which may not fully reflect long-term graft survival or functional recovery. The small perfusate volume and extended perfusion duration also increased the risk of rapid substrate depletion, a known challenge in NP circuits that can alter metabolic profiles over time [22]. Another limitation is the relatively small number of animals in each group, which restricts the generalizability of our findings. Larger-scale studies will be necessary to validate these preliminary findings.

## 5. Conclusions

In conclusion, elevated initial mean perfusion pressure negatively affects donor heart preservation during NP, contributing to metabolic derangement, myocardial injury, and eventual graft dysfunction. These findings underscore the importance of carefully regulating perfusion pressure to optimize graft viability and support successful transplantation outcomes.

## Figures and Tables

**Figure 1 medicina-61-01696-f001:**
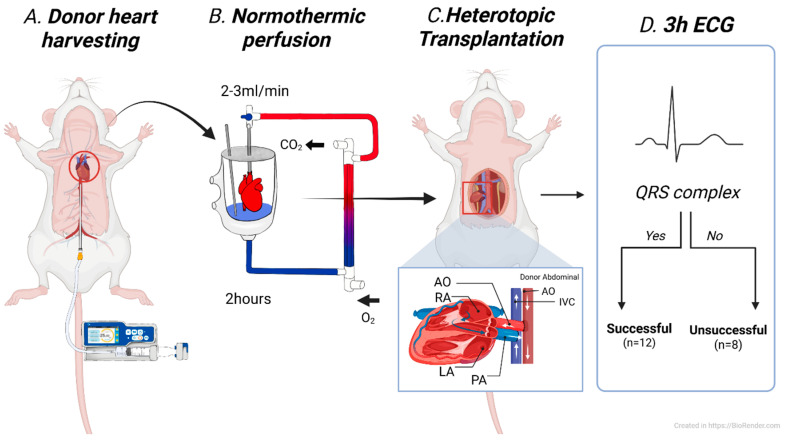
Experimental study design, AO (Aorta), RA (Right Atrium), LA (Left Atrium), PA (Pulmonary Artery), QRS (QRS complex), and ECG (Electrocardiogram).

**Figure 2 medicina-61-01696-f002:**
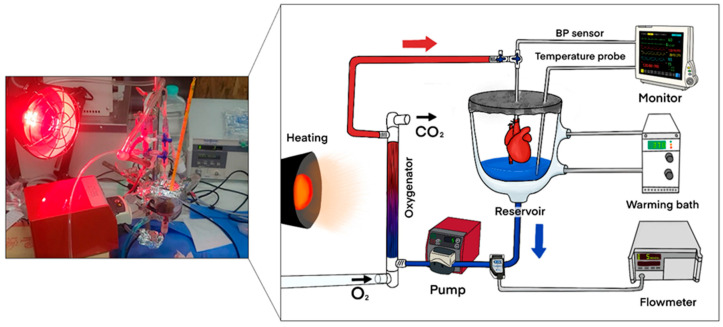
Schematic diagram of the normothermic ex situ perfusion circuit used in this study. The system consisted of a pump, oxygenator, reservoir, flowmeter, warming bath, and monitoring devices (blood pressure, temperature, and electrocardiography).

**Figure 3 medicina-61-01696-f003:**
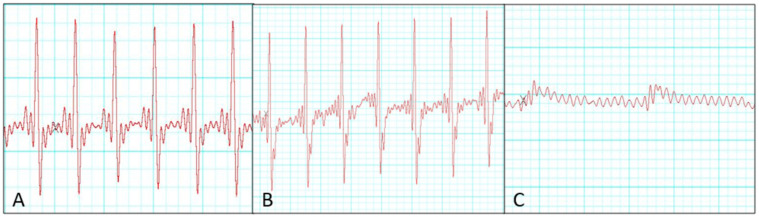
The donor hearts were classified based on the ECG examination after 2 h of NP preservation. (**A**) ECG before donor heart harvesting, (**B**) ECG after successful NP preservation, and (**C**) ECG after unsuccessful NP preservation.

**Figure 4 medicina-61-01696-f004:**
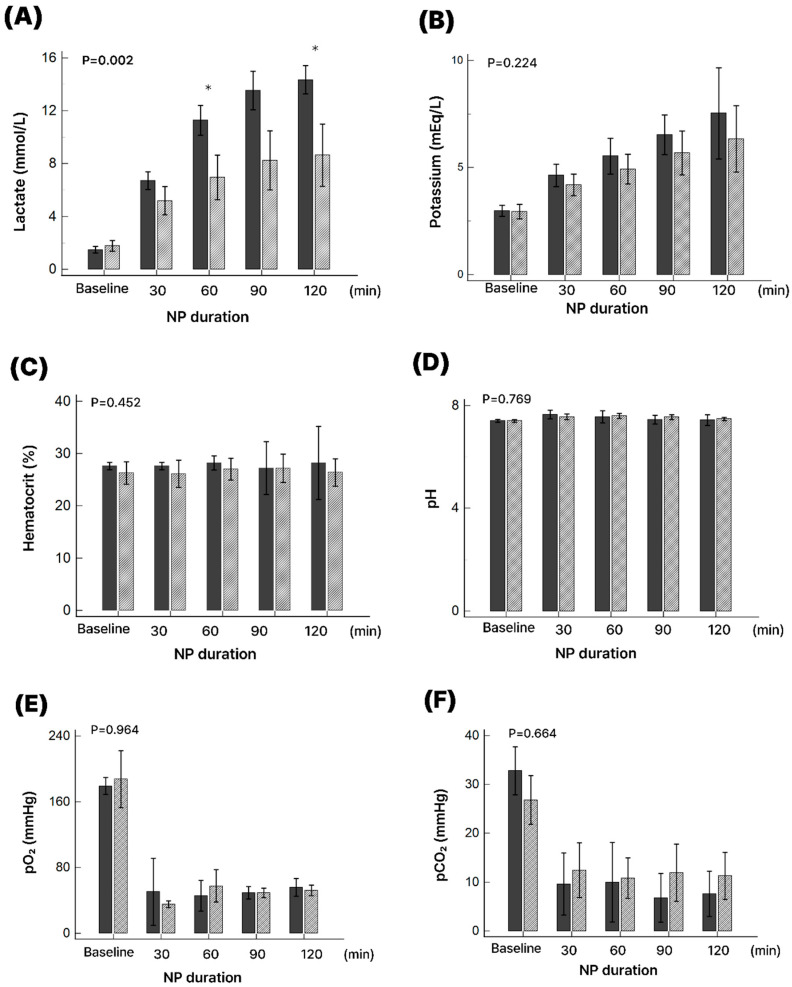
Serial changes in perfusion parameters during normothermic ex situ perfusion. (**A**) Lactate (*p* = 0.002), (**B**) potassium (*p* = 0.224), (**C**) hematocrit (*p* = 0.452), (**D**) pH (*p* = 0.769), (**E**) pO_2_ (*p* = 0.964), and (**F**) pCO_2_ (*p* = 0.664). Solid bars represent the unsuccessful group, and diagonal-hatched bars represent the successful group. An asterisk (*) denotes statistical significance.

**Figure 5 medicina-61-01696-f005:**
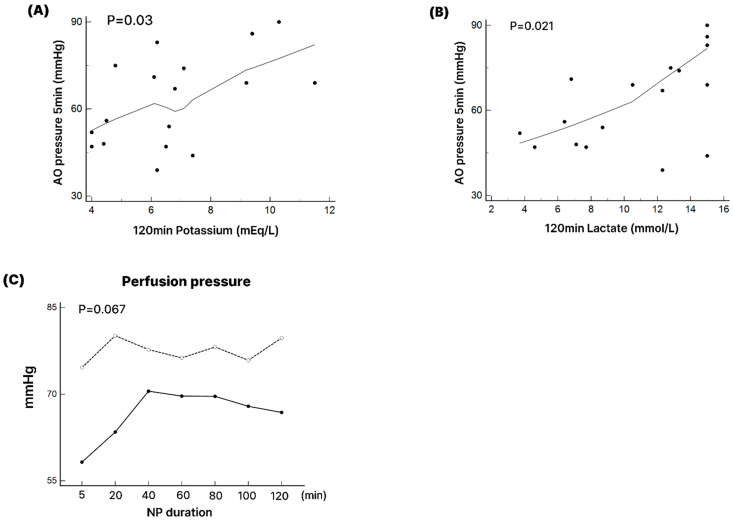
Correlation analysis to determine the factors associated with failure. (**A**,**B**) Correlation between initial perfusion pressure and the level of potassium (**A**) and lactate (**B**) after two hours of NP preservation. (**C**) Changes in the perfusion pressure between successful and unsuccessful groups during the 2 h of NP preservation. Ao. Pr.—perfusion pressure. Open circles and dotted line represent the unsuccessful group, whereas filled circles and solid line represent the successful group. Correlation coefficient analysis: potassium (*p* = 0.03), lactate (*p* = 0.021), and perfusion pressure (*p* = 0.067).

**Figure 6 medicina-61-01696-f006:**
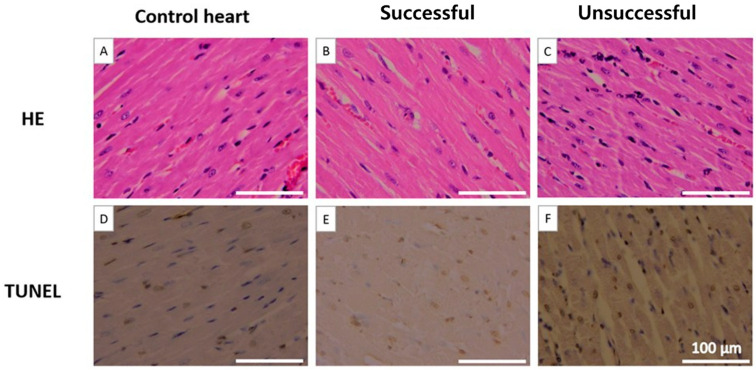
Histological and TUNEL staining of donor hearts after 4 h of heterotopic transplantation. (**A**) Control heart with preserved myocardial structure (H&E). (**B**) Successful group showing well-preserved myocardial fibers (H&E). (**C**) Unsuccessful group with irregular cardiomyocytes, interstitial edema, and inflammatory cell infiltration (H&E). (**D**) Control heart without apoptotic nuclei (TUNEL). (**E**) Successful group with few TUNEL-positive nuclei. (**F**) Unsuccessful group with abundant TUNEL-positive nuclei. Overall, the successful group preserved myocardial integrity and exhibited fewer apoptotic changes compared with the unsuccessful group. Scale bars = 100 μm.

**Figure 7 medicina-61-01696-f007:**
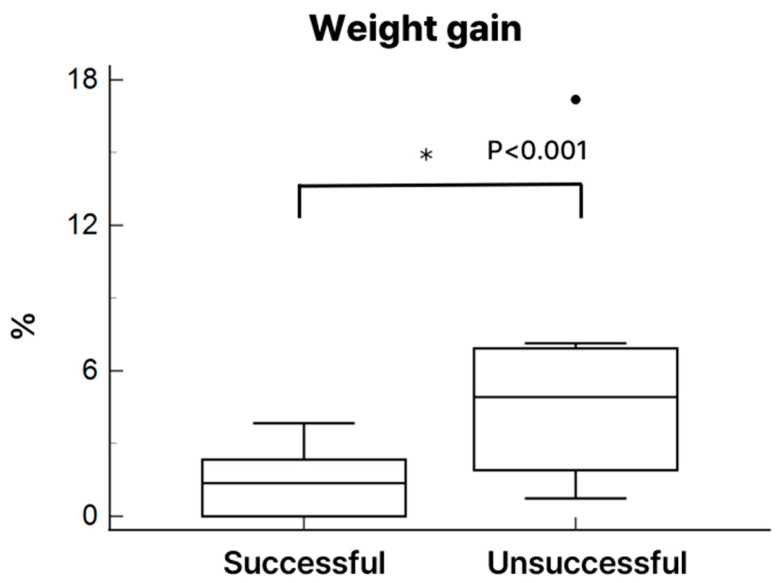
The change in the heart weight after NP preservation. *, *p* < 0.01.

**Table 1 medicina-61-01696-t001:** Comparison of ischemic time between successful and unsuccessful transplantation groups (mean ± SD).

	Successful	Unsuccessful	*p*-Value
**Donor and Recipient**			
Recipient clamping time (min)	45.4 ± 6.2	47.9 ± 3.9	0.36
Donor heart			
Harvesting time (min)	12.7 ± 1.3	13.6 ± 1.6	0.19
Total ischemic time (min)	46.2 ± 7.4	46.8 ± 2.4	0.83
Total out-of-body time (min)	168.3 ± 8.6	166.8 ± 2.4	0.66

## Data Availability

The data and material generated during and analyzed in this study are available from the corresponding author on reasonable request.

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
