# Peer review of "Optimal Perfusion Pressure Enhances Donor Heart Preservation During Normothermic Ex Situ Perfusion in a Rat Transplantation Model"

_medicina, 2025, doi:10.3390/medicina61091696_

Round 1

Reviewer 1 Report

Comments and Suggestions for Authors

The paper by Do Wan Kim et al., titled ‘Optimal perfusion pressure enhances donor heart preservation during normothermic ex-situ perfusion in a rat transplantation model’, is a well-written, original manuscript that addresses an important issue: the preservation of donor hearts.

However, before the publication, some minor concerns should be resolved.

The Materials and Methods section should be restructured to present the work in a step-by-step manner, as illustrated in Figure 1. I would advise starting with the source and description of the animals. Please consider changing ‘Classification’ to ‘Study Groups’, for instance. Classification does not seem a suitable title for this paragraph. I would also recommend shortening the section.   

Figure 1. – Check spelling of ‘heterotopic’.

Lines 193-194: Obviously, this should be Table 1, not Figure 1.

Results section.

The first two sentences in the Results section repeat the information from the Materials and Methods section. I would recommend deleting them.

Line 157. The authors cited Table 1 rather than Figure 1.

Please check the message referring to Fig. 2 (lines 160-161). The Fig.2 illustrates electrical activity rather than heart rate and should be cited at the beginning of the ‘Results’ section.

Change the name, labels, etc. on Fig.3.

Line 174: Consider changing the title of this paragraph, as it does describe the results of a comparative analysis.

Line 227: Figure 4. Correlation analysis to determine the factors leading to failure.

It would be more accurate to change ‘leading’ to ‘associated’.

Lines 179-181. Please provide the calculations or explain how the value of 65mmHg was chosen as the cut-off point.

Fig.5: Instead of Gr.1 and Gr.2, it would be better to use a single name for the study groups: successful and unsuccessful.

Fig.6: Please check and clarify the difference between the two parts of this Figure.

Discussion section.

Lines 268-275: from this paragraph, it becomes clear that ECG has not previously been used to assess heart transplant preservation. If so, the authors should explain why they chose ECG as the method for dividing the study group.

Author Response

Thank you for the valuable suggestion and for taking the time to review our manuscript. We carefully considered your comments and revised the manuscript accordingly.

To clearly indicate the revisions:

  • Added or newly revised sentences are highlighted in yellow.
  • Replaced words within sentences are marked in red.
  • Deleted words or phrases are shown with a

We have addressed each of your comments point by point as follows:

Comment 1: The Materials and Methods section should be restructured to present the work in a step-by-step manner, as illustrated in Figure 1. I would advise starting with the source and description of the animals. Please consider changing ‘Classification’ to ‘Study Groups’, for instance. Classification does not seem a suitable title for this paragraph. I would also recommend shortening the section.  

Response 1: The section has been reorganized step by step to follow the flow of Figure 1, and the heading “Classification” has been changed to “Study Groups” to enhance clarity.

Comment 2: Figure 1. – Check spelling of ‘heterotopic’.

Response 2: The typographical error has been corrected.

Comment 3: Lines 193-194: Obviously, this should be Table 1, not Figure 1.

Response 3: The citation has been corrected to Table 1 instead of Figure 1 in lines 193–194.

Results section.

Comment 4: The first two sentences in the Results section repeat the information from the Materials and Methods section. I would recommend deleting them.

Response 4: The duplicated first sentence has been deleted, and a new subheading ((1) Ischemic time and electrocardiographic change) was added. In addition, the paragraph was streamlined and reorganized to improve readability.

Comment 5: Line 157. The authors cited Table 1 rather than Figure 1.

Response 5: The citation has been corrected to Table 1 instead of Figure 1 in line 157.

Comment 6: Please check the message referring to Fig. 2 (lines 160-161). The Fig.2 illustrates electrical activity rather than heart rate and should be cited at the beginning of the ‘Results’ section.

Response 6: The legend of Figure 2 and the related description in the main text have been revised to clarify that it represents electrical activity rather than heart rate (highlighted in red).

Comment 7: Change the name, labels, etc. on Fig.3.

Response 7: The title and axis labels of Figure 3 have been revised for greater clarity.

Comment 8: Line 174: Consider changing the title of this paragraph, as it does describe the results of a comparative analysis.

Response 8: The subheading in line 174 has been modified to better reflect the comparative analysis presented in the section.

Comment 9: Line 227: Figure 4. Correlation analysis to determine the factors leading to failure.

It would be more accurate to change ‘leading’ to ‘associated’.

Response 9: Following the suggestion, “leading” was replaced with “associated” to emphasize correlation rather than causation in the Figure 4 caption.

Comment 10: Lines 179-181. Please provide the calculations or explain how the value of 65mmHg was chosen as the cut-off point.

Response 10: The value of 65 mmHg was identified as a threshold where lactate and potassium levels significantly increased in the failure group. Accordingly, 65 mmHg was adopted as the reference value in this study, and the explanation was added to the last sentence of Results (3).

Comment 11: Fig.5: Instead of Gr.1 and Gr.2, it would be better to use a single name for the study groups: successful and unsuccessful.

Response 11: The group labels have been revised to “successful” and “unsuccessful” as suggested.

Comment 12: Fig.6: Please check and clarify the difference between the two parts of this Figure.

Response 12: Two overlapping graphs were removed in Figure 6 to make the differences between components more evident.

Discussion section.

Comment 13: Lines 268-275: from this paragraph, it becomes clear that ECG has not previously been used to assess heart transplant preservation. If so, the authors should explain why they chose ECG as the method for dividing the study group.

Response 13: ECG was selected as the group classification criterion because the presence of the QRS complex directly reflects myocardial electrical activity, which is a key indicator of graft viability. Compared to metabolic and hemodynamic markers, ECG provides immediate and reliable information, making it the most suitable parameter for our model. This rationale has been reinforced in the third sentence of the fourth paragraph in the Discussion.

Reviewer 2 Report

Comments and Suggestions for Authors

Dear Authors,

I wish to congratulate you on the submitted manuscript entitled: "Optimal perfusion pressure enhances donor heart preservation during normothermic ex-situ perfusion in a rat transplantation model". The submitted manuscript is very interesting and revealing.

1. Your study aimed to identify the optimal perfusion pressure to enhance graft preservation in rat heart transplantation, as twenty male Sprague- Dawley rats were used. According to your protocol, donor hearts underwent normothermic preservation for 2 hours using a Langendorff apparatus primed with 12 mL of solution at a consistent 3 mL/min flow. My question that arose while I was reading your manuscript is why you did not use Mannitol solution as an additive to the perfusate. From a practical point of view, Mannitol addition was required, justified, and could change the presented results profoundly. 

2. As you mentioned, higher initial perfusion pressures (>65 mmHg) at a constant flow rate resulted in elevated lactate and potassium concentrations post-preservation, indicating suboptimal outcomes. Was the high potassium the primary reason for the QRS absence? The information should be clearly presented, discussed, and potential therapeutic steps suggested.  

3. As the number of the involved animals in both groups did not allow for general conclusion drawing, the information should be added and discussed. 

I wish to congratulate you, once again, on the submitted manuscript.

kinds

R

Author Response

Thank you for the valuable suggestion and for taking the time to review our manuscript. We carefully considered your comments and revised the manuscript accordingly.

To clearly indicate the revisions:

  • Added or newly revised sentences are highlighted in yellow.
  • Replaced words within sentences are marked in red.
  • Deleted words or phrases are shown with a

We have addressed each of your comments point by point as follows:

Comment 1: 1. Your study aimed to identify the optimal perfusion pressure to enhance graft preservation in rat heart transplantation, as twenty male Sprague- Dawley rats were used. According to your protocol, donor hearts underwent normothermic preservation for 2 hours using a Langendorff apparatus primed with 12 mL of solution at a consistent 3 mL/min flow. My question that arose while I was reading your manuscript is why you did not use Mannitol solution as an additive to the perfusate. From a practical point of view, Mannitol addition was required, justified, and could change the presented results profoundly.

Response 1: In this study, we prioritized the basic composition used in a previously established rat Langendorff perfusion model (Rat Model of Normothermic Ex-Situ Perfused Heterotopic Heart Transplantation, DOI: 10.3791/64954), which included perfusate with autologous blood, saline, glucose, insulin, and antibiotics, without mannitol. However, as the reviewer pointed out, mannitol, as an osmotic component, can reduce cellular edema and oxidative stress, thereby exerting a positive effect on myocardial preservation. Since the primary aim of this study was to investigate metabolic and electrophysiological changes according to perfusion pressure, mannitol was not added. Nevertheless, we will incorporate comparative analyses of different perfusate compositions, including mannitol, in future studies.

Comment 2: As you mentioned, higher initial perfusion pressures (>65 mmHg) at a constant flow rate resulted in elevated lactate and potassium concentrations post-preservation, indicating suboptimal outcomes. Was the high potassium the primary reason for the QRS absence? The information should be clearly presented, discussed, and potential therapeutic steps suggested. 

Response 2: In our results, the unsuccessful group exhibited elevated perfusion pressure, increased potassium levels, and loss of the QRS complex. Potassium accumulation may have contributed to conduction abnormalities, and potential therapeutic approaches to address this issue include optimizing the electrolyte composition of the perfusate and regulating potassium levels. This explanation has been added to the second sentence of the fourth paragraph in the Discussion.

Comment 3: As the number of the involved animals in both groups did not allow for general conclusion drawing, the information should be added and discussed.

Response 3: This study was designed as a preclinical exploratory investigation, and the small number of animals in each group poses a limitation to generalizability. This has been clearly stated as a limitation in the last sentence of the sixth paragraph in the Discussion so that readers may take it into account when interpreting the results.

Round 2

Reviewer 2 Report

Comments and Suggestions for Authors

I have no further comments.